# Transcriptome-Wide Analysis Reveals Key DEGs in Flower Color Regulation of *Hosta plantaginea* (Lam.) Aschers

**DOI:** 10.3390/genes11010031

**Published:** 2019-12-26

**Authors:** Jingying Zhang, Changhai Sui, Yanli Wang, Shuying Liu, Huimin Liu, Zhengren Zhang, Hongzhang Liu

**Affiliations:** 1College of Life Sciences, Jilin Agricultural University, Changchun 130118, China; zhangjingying@jlau.edu.cn (J.Z.); suich.cn@gmail.com (C.S.); wangyanli@jlau.edu.cn (Y.W.); liushuyingbr@jlau.edu.cn (S.L.); Liuhuimin@jlau.edu.cn (H.L.); zhangzhengren@jlau.edu.cn (Z.Z.); 2Department of bioengineering, Jilin Engineering Vocational College, Siping 136001, China

**Keywords:** *Hosta plantaginea* (Lam.) Aschers, transcriptome, flower colors, differentially expressed genes (DEGs)

## Abstract

Background: *Hosta plantaginea* (Lam.) Aschers (HPA), a species in the family Liliaceae, is an important landscaping plant and herbaceous ornamental flower. However, because the flower has only two colors, white and purple, color matching applications are extremely limited. To date, the mechanism underlying flower color regulation remains unclear. Methods: In this study, the transcriptomes of three cultivars—*H. plantaginea* (HP, white flower), *H. Cathayana* (HC, purple flower), and *H. plantaginea ‘Summer Fragrance’* (HS, purple flower)—at three flowering stages (bud stage, initial stage, and late flowering stage) were sequenced with the Illumina HiSeq 2000 (San Diego, CA, USA). The RNA-Seq results were validated by qRT-PCR of eight differentially expressed genes (DEGs). Then, we further analyzed the relationship between anthocyanidin synthase (ANS), chalcone synthase (CHS), and P450 and the flower color regulation by Gene Ontology (GO), Kyoto Encyclopedia of Genes and Genomes (KEGG), and Eukaryotic Orthologous Groups (KOG) network and pathway enrichment analyses. The overexpression of CHS and ANS in transgenic tobacco petals was verified using qRT-PCR, and the petal colors associated with the overexpression lines were confirmed using absorbance values. Results: Over 434,349 transcripts were isolated, and 302,832 unigenes were identified. Additionally, through transcriptome comparisons, 2098, 722, and 606 DEGs between the different stages were found for HP, HC, and HS, respectively. Furthermore, GO and KEGG pathway analyses showed that 84 color-related DEGs were enriched in 22 pathways. In particular, the flavonoid biosynthetic pathway, regulated by *CHS*, *ANS*, and the cytochrome P450-type monooxygenase gene, was upregulated in both purple flower varieties in the late flowering stage. In contrast, this gene was hardly expressed in the white flower variety, which was verified in the *CHS* and *ANS* overexpression transgenic tobacco petals. Conclusions: The results suggest that CHS, ANS, and the cytochrome P450s-regulated flavonoid biosynthetic pathway might play key roles in the regulation of flower color in HPA. These insights into the mechanism of flower color regulation could be used to guide artificial breeding of polychrome varieties of ornamental flowers.

## 1. Introduction

*Hosta plantaginea* (Lam.) Aschers (HPA) is a perennial herb of the Liliaceae family and native to Southeast China, Japan, the Korean Peninsula, and Far East Russia. HPA is an economically valuable herb: it has been cultured for more than 2000 years in China as an important landscaping plant and herbal ornamental flower, and it is used as a traditional Chinese medicine, as recorded in the “Compendium of Materia Medica” and other ancient medical books. It is mainly distributed in the temperate and subtropical forest margins, under the forest, and on the waterside of East Asia. It is a typical shade-tolerant mosaic. The genus *Hosta* includes 3 subgenera, 10 groups, 43 species, and 35 varieties (or variants), of which *H. plantaginea*, *H. ventricosa*, *H. ensata*, and *H. albofarinosa* are native to China [1].

Flower color, an important quality indicator of ornamental plants that directly determines their ornamental value, is determined by pigment composition and content, pH value, petal epidermal cellular shape, and external environmental factors, among others [2,3,4]. The anthocyanin biosynthesis pathway can be divided into three stages (Appendix A). To date, four major compound types-flavonoids, chlorophyll, carotenoids, water-soluble alkaloids, and their derivatives-that contribute to flower coloring have been separated and purified [5,6]. Flavonoid compounds play pivotal roles in flower color and produce a continuous spectrum from light yellow to blue-violet. Flavonoids are known to be associated with the metabolism and synthesis of anthocyanins and determine the final color of flowers [7,8,9]. Anthocyanins are water-soluble pigments found in almost all vascular plants and can change the flower color into white, yellow, orange, red, purple, blue, and other colors [5,8]. As reported by Tanaka [5,10], the biosynthetic pathway of flavonoids has been well established, and the key genes (such as cytochrome P450 and glucosyltransferase) have been determined in other flowers and plants [11,12,13].

However, wild-type HPA is found with only two flower colors (purple and white), and the molecular mechanism that determines flower color remains unclear. In this study, we analyzed the transcriptomes of different flowering stages of three species of HPA using Illumina HiSeq 2000 (San Diego, CA, USA) technology. Subsequently, the functions and pathways associated with the differentially expressed genes (DEGs), as determined by RNA-Seq and validated by qRT-PCR, were analyzed using Gene Ontology (GO), Kyoto Encyclopedia of Genes and Genomes (KEGG), and Eukaryotic Orthologous Groups (KOG) enrichment analyses. Chalcone synthase (CHS) and ANS were overexpressed in transgenic tobacco petals, and the relationships between the expression levels of ANS, CHS, and P450 and flower color were analyzed using qRT-PCR and absorbance values. Our results provide a better understanding of the transcriptomic changes and the underlying molecular mechanism of flower color in HPA.

## 2. Materials and Methods

### 2.1. Plant Materials and Growth Conditions

The experimental materials were *H. plantaginea* (HP), *H. Cathayana* (HC), and *H. plantaginea ‘Summer Fragrance’* (HS), which were selected to represent different cultivars of HPA. The plants were cultivated in the Botanical Garden of the Institute of Botany, Chinese Academy of Sciences (east longitude 116°13′9.4116″, north latitude 39°59′55.8312″; 67 m above sea level). The cultivation substrate was garden soil: yellow-sand soil: perlite = 3:1:1, which contained total nitrogen 0.012%, including hydrolysable nitrogen 11.3 mg/kg, total phosphorus 12.2 mg/kg, total potassium 1.68%, organic material 0.312%, pH = 8.2. These plants are mainly distributed in temperate and subtropical forest margins, under the forest canopy and along the waterside of East Asia. They are typical shade tolerant mosaics. During the flowering period (from June to September), the flowers were collected at the bud stage, initial stage, and late flowering stage. The collected flowers were stored in an ultra-cold storage freezer at −80 °C.

### 2.2. RNA Extraction and Sequencing

Total RNA from each sample, which had a 0.1 g sample weight from the three different flowering stages of the three varieties HP, HS, HC, was extracted using TRIzol^®^ Reagent (Life Technologies Corporation, Carlsbad, CA, USA) and treated with NEBNext^®^ DNase I (New England Biolabs, Ipswich, MA, USA) according to the manufacturer’s protocol. Poly(A) mRNA was enriched using oligo (dT) beads and fragmented using fragmentation buffer. Finally, 100 ng of purified and enriched mRNA was used to construct a cDNA library for each sample using NEBNext^®^ Ultra™ RNA Library Prep Kit for Illumina (New England Biolabs, Ipswich, MA, USA). cDNA fragments of 200 bp (±25 bp) were selected and purified by gel electrophoresis and used as templates for amplification with PCR for end-repair and poly(A) addition. The purified library products were evaluated using the Agilent 2200 Tape Station and Qubit 2.0 software (Life Technologies, Carlsbad, CA, USA). Trinity software was used for transcriptome assembly using the default software parameters. RNA-Seq was performed at Novogene Science and Technology Co. Ltd. (Beijing, China) using the HiSeq™ 2000 (Illumina, San Diego, CA, USA). The experimental data were obtained from three independent experiments.

### 2.3. Transcriptome Assembly

All subsequent analyses were performed using clean data [14,15,16,17,18]. The clean reads were assembled de novo into longer contigs on the basis of overlapping regions using the Trinity platform (http://trinityrnaseq.sourceforge.net/). The gene expression levels were indicated as FPKM (Fragments Per Kilobase of exon per Million fragments mapped). Genes in the three different samples with a q-value ≤0.05 and a fold change ≥2 were considered to be significantly differentially expressed genes [19]. The raw image files were obtained and analyzed to render sequenced reads by CASAVA base calling using Illumina HiSeq™ after visual evaluation of the sequencing data by FastQC. The Phred quality score (Q) is an integer mapping of the probability (P) of an incorrect base call, with a mapping relation of Q = −10 log(P). The clean de novo data were assembled into transcripts using Trinity. The main parameter was min_kmer_cov 2, and the rest of the parameter settings were set to the default values.

### 2.4. Validation of Gene Expression with qRT-PCR

The samples for PCR validation were selected from the same batch of petals used for sequencing, and total RNA was used to synthesize cDNA with the PrimeScript™ RT reagent Kit (Perfect Real Time, TaKaRa, Kusatsu, Japan). The HPA housekeeping gene actin3 was used as an internal control for normalization. Primers for qRT-PCR of eight DEGs were designed with Premier 6.0 software and are shown in Appendix A. qRT-PCR was performed using the SYBR Green One-Step qRT-PCR kit (TransGen Biotech, Beijing, China). Three biological replicates of three independent experiments were performed per sample.

### 2.5. Gene Ontology Annotation

All expressed genes in the obtained transcriptomes were annotated on the basis of BLAST homology searches and searched against the Swiss-Prot and TrEMBL databases by double-direction BLAST [20,21,22]. Functional annotation (GO terms) were downloaded from the UniProt database (http://www.uniprot.org/uniprot). For GO enrichment analysis, the DEGs were mapped to GO terms in the GO database (http://www.geneontology.org) to retrieve GO annotations for each DEG. For the KEGG pathway analysis, KEGG orthology terms were assigned to DEGs from the KEGG pathway database (http://www.genome.jp/kegg/). The functions of DEGs in different samples were further explored by performing KEGG enrichment analyses (*p* < 0.01, FDR < 0.05) using hypergeometric tests with Blastall software. Heatmaps were drawn using the Pheatmap package in R (https://cran.r-project.org/package1/4pheatmap) [21,22,23].

### 2.6. Investigation of Flower Color Regulation by CHS and ANS Genes in Tobacco

To verify the importance of the *CHS* and *ANS* in the regulation of petal color, we prepared transgenic tobacco by the *Agrobacterium* method and analyzed the color of transgenic tobacco leaves. Seeds of wild-type tobacco NC89, *Agrobacterium tumefaciens* EHA105, and the expression vector pCAMBIA3301 (Appendix A) were provided by the Changbai Mountain Characteristic Plant Resources Research Laboratory of Jilin Agricultural University. The experiment was performed as previously described [24,25]. Briefly, tobacco leaves with wounds of about 1 cm^2^ were infected with *Agrobacterium* solution containing recombinant EC2.3.1.74 (CHS) and EC1.14.11.19 (ANS) plant vectors and then incubated in co-culture medium for 3 days. After 30 days of screening culture, the buds on the callus were cut, differentiated for 15 days, and then cultured in rooting medium for 20 days. The well-rooted tobacco was transplanted to the soil and cultivated until the tobacco blossomed and set seed. We had collected three independent transgenic tobacco petals to extract RNA. The overexpression of CHS and ANS in transgenic tobacco petals was verified by qRT-PCR. The total anthocyanin content in tobacco petals in 1% methanol hydrochloric acid extract was determined by measuring the absorbance values at 530 and 657 nm using an ultraviolet spectrophotometer. The relative content of total anthocyanins was calculated as (A530 − 0.25 × A657)/Sample Weight [26].

## 3. Results

### 3.1. Flower Colors of HPA at Different Developmental Stages

Three cultivars of the HPA species, HP (white flower), HC (purple flower), and HS (purple flower), were selected as the experimental materials. The flowers of HC and HS were purple at the bud stage, and the color gradually deepened during the initial stage and late flowering stage. In contrast, HP exhibited a significant difference in flower color and remained white during all three stages (Figure 1).

### 3.2. Transcriptome Sequencing and Assembly

Table 1 shows the nine transcriptome libraries constructed for HC, HP, and HS at each of the three stages. More than 98% high-quality clean reads were identified after the visual evaluation of sequencing data by FastQC. In total, 302,832 unigenes and 434,349 transcripts were detected and assembled.

### 3.3. Analyses of DEGs

From the Illumina data, we set *p*-value < 0.05 and |FoldChange| > 2 to identify DEGs from the heatmaps of the three flower petal varieties of HPA (Figure 2A). Through multiple transcriptome comparisons, 11,854 DEGs were obtained for the three flowering stages of the three varieties. Furthermore, we discovered 2098 DEGs between the three stages of HP, 722 DEGs between the three stages of HC and 606 DEGs between the three stages of HS. The DEG data in different flowering phases were compared and analyzed to construct a histogram (Figure 2B), Venn diagram (Figure 2C), and heatmap (Figure 2D). From volcano map (Appendix A) and heatmap analyses, 17,845 DEGs between HP-3 and HS-3 were obtained, with 5858 upregulated and 11,987 downregulated DEGs in HS-3. On the other side, 21,887 DEGs were obtained between HP-3 and HC-3, with 9097 upregulated and 12,790 downregulated DEGs in HC-3. The results of all analyses include the genes of interest: *CHS*, *ANS*, and *P450-type monooxygenase*. The Venn diagram further showed that the samples in the initial flowering period and the flowering period have 450 DEGs in common that include a higher number of flavonoid biosynthesis pathway genes, such as the *CHS*, *ANS*, and *P450-type monooxygenase* genes.

### 3.4. Validation of DEGs by qRT-PCR

The expression levels of eight DEGs were validated using qRT-PCR. The qRT-PCR results showed that the DEG expression levels were consistent with those obtained using RNA-Seq, which confirms that the transcriptome data were correct and that the RNA-Seq process was effective (Figure 3A). Specifically, in the petal color regulation process, the expression levels of ANS, CHS, and P450 genes were higher in the late flowering stage than in the bud stage and initial stage (Figure 3B), suggesting that these genes play a key role in color regulation.

### 3.5. Pathway and Functional Analysis of DEGs

To analyze the DEGs in the flowering stage of the three samples (i.e., HP-3, HS-3, and HC-3), we researched the distribution of the DEGs using GO, KEGG, and KOG annotations. Of the 434,349 transcripts identified between HP-3 vs. HC-3 and HS-3, we obtained annotations for 32,556 DEGs and information for 3210 proteins. In the ‘Molecular Function GO category, we obtained 13,997 DEGs for HP-3 vs. HC-3 (205 DEGs were enriched in enzyme regulator activity (GO: 0030234); 6515 DEGs were enriched in catalytic activity (GO: 0003824)), and 16,440 DEGs for HP-3 vs. HS-3 (164 DEGs were enriched in GO: 0030234; 5498 DEGs were enriched in GO: 0003824). The KEGG enrichment analysis from the obtained GO data showed 19 KEGG Orthology Identifiers (KO IDs) for HP-3 vs. HC-3 (Figure 4A1) and 20 KO IDs for HP-3 vs. HS-3 (Figure 4A2). In the scatter maps, the smaller the q-value, the closer the color is to red, and the size of the points represent the number of DEGs enriched in the corresponding function. The functional interaction network from the KEGG enrichment results was used to identify the functional sites of xenobiotics metabolism by cytochrome P450 in HP-3 vs. HC-3 (Figure 4B1). Six sites were associated with cytochrome P450 metabolism in HP-3 vs. HS-3 (Figure 4B2). The color of the nodes represents the enrichment degree of the function (as the q-value). The higher the enrichment degree, the lower the *p*-value and the darker the color.

After annotating the functions of DEGs, the metabolic pathways involving the DEGs were obtained by the statistical analysis of the GO, KEGG, and KOG enrichment results (Appendix A). The pathway map in Figure 5A shows the pathway for the metabolism of xenobiotics by cytochrome P450, which involves CHS, ANS, and the cytochrome P450 protein, which plays a key role in the regulation of color expression. The top five GO terms were analyzed using the Directed Acyclic Graph (DAG) of biological processes (Figure 5B) and molecular function (Figure 5C).

### 3.6. Study on Flower Color Regulation of CHS and ANS Genes in Tobacco

The flowers of tobacco plants that were positive for the overexpression of *CHS* and *ANS* genes were examined, and the results revealed that the flower color of the overexpressing transgenic tobacco plants was deepened compared with wild-type tobacco flowers (Figure 6A,B). qRT-PCR confirmed that the expression levels of the *CHS* and *ANS* genes in transgenic tobacco petals were six-fold higher than those of the wild type (Figure 6C,D). As determined by the absorbance values of a 1% hydrochloric acid and methanol extraction of tobacco flowers, the total anthocyanin content in the *CHS*-overexpressing tobacco was about three-fold higher than that of the wild type, and the total anthocyanin content in the *ANS*-overexpressing tobacco was two times that of the wild type (Figure 6E,F). Thus, the overexpression of the *CHS* and *ANS* genes significantly increased the expression of flower color.

## 4. Discussions

HPA has been cultured for 2000 years in China as an important landscaping plant and ornamental herbal flower. Furthermore, according to “Compendium of Materia Medica”, its medicinal functions include detumescence, detoxification, and hemostasis. Flower color is an important quality index of ornamental plants, and it directly determines the visual value of plants. The narrow definition of flower color refers to the color of petals, and the broad definition includes not only the color of petals of flowering plants but also the color of their reproductive organs, such as the pistil and stamen. To date, four major leaf lines (green, blue-gray, yellow, and polychrome) and more than 5000 internationally registered varieties have been bred after long-term cultivation and cross-breeding. However, HPA flowers are only found in two colors, white and purple, and only a few varieties have any color. Therefore, the garden and ornamental applications of HPA are limited [1].

Forkmann [6] divided flower-color-related proteins into seven categories: (1) involved in a single step in the biosynthesis of flavonoids, (2) modify related genes after flavonoid synthesis, (3) regulatory genes that open or close a whole synthetic pathway or part of it, (4) genes and factors that affect the flavonoid concentration, (5) proteins and factors related to flower structure, (6) proteins and factors that affect color, and (7) control flower morphology. Since the middle of the 19th century, according to their chemical structure, cell location, and biochemical synthesis pathway, pigments have been classified as carotenoids, alkaloids, and flavonoids [5,27]. In terms of chemical structure, flavonoids are secondary plant metabolites and represent a series of compounds whose primary chemical structure is 2-phenyl chromogenic ketone. Anthocyanins are water-soluble pigments found in almost all vascular plants, and they can produce white, yellow, orange, red, purple, blue, and other colors in flowers [5,28]. The basic structure of flower color is 3,5, 7-trihydroxy-2-phenyl benzopyran, and the main structure of anthocyanin components in *Hosta* spp. contains a glucoside; however, researchers have suggested that the single color of *Hosta* spp. is decided by the dominant synthesis of malvidin 3,5-diglucoside and malvidin 3-coumaroyl glucoside 7-glucoside [29,30,31].

In this study, we investigated the changes in flower color of three native, cultivated varieties of HPA in China. HP, with white flowers, and HC and HS, both with purple flowers, were found to be significantly different in color. Transcriptome analysis was performed using Illumina HiSeq 2000 (San Diego, CA, USA) to identify the DEGs and explain the mechanism that controls flower color. We obtained 302,832 unigenes, and 434,349 transcripts were assembled from samples collected during the bud stage, initial stage, and late flowering stage. In total, 11,854 DEGs were obtained, and the greatest number of DEGs were found in HP (2098), followed by HC (722) and then HS (606) (Figure 2A). Furthermore, 84 color-related DEGs between purple and white samples were identified, and they are associated with anthocyanins, flavonoids, and flavonols, which contribute to the ranges of flower color from red to purple and determine the final flower colors [4,7].

Gene enrichment analysis was performed to analyze the association of the expression levels of the DEGs with KOG, GO, and KEGG results. More than 32,556 DEGs were annotated, and information was obtained for 3210 proteins. They were determined to belong to more than 1687 GO terms and enriched in 1551 GO terms (Chromatin structure and protein interactions, General function prediction only, and Signal transduction mechanisms) (Figure 4A). Moreover, the KEGG and KOG analysis found 7902 genes that encode enzymes (Figure 4) and were enriched in 11 pathways: the metabolism pathway of flavonoids by cytochrome P450s, metabolic pathways, the biosynthesis of secondary metabolites, microbial metabolism in diverse environments, the biosynthesis of antibiotics, carbon metabolism, plant hormone signal transduction, phenylalanine metabolism, arginine and proline metabolism, glycerophospholipid metabolism, and steroid hormone biosynthesis. The *P450* gene of c120890 suggests that direct regulation of downstream anthocyanin-related genes occurs via the metabolic pathway of flavonoids by cytochrome P450s.

Flavonoids, a major class of plant secondary metabolites, are found in most plants and involved in plant growth, development, and protection. Our study showed that the metabolism of xenobiotics by the cytochrome P450 pathway and flavonoid biosynthesis were enriched in the metabolic pathway of flavonoids by CHS, ANS, and cytochrome P450s. ANS is an α-ketoglutarate-dependent dioxygenase that is located downstream of the anthocyanin biosynthesis pathway and catalyzes the color anthocyanin formation by α-ketoglutaric acid and Fe^2+^ [32]. In plants, cytochrome P450s mediate hydroxylation processes, such as the formation of heteroatoms, dealkylation, deamination, dehalogenation, and epoxidation at nitrogen and sulfur sites [33,34]. Moreover, cytochrome P450 enzymes induce the hydroxylation of dihydroflavonols at the 3′ and 5′ positions, which is an important step that determines whether purple-blue or red anthocyanins are formed. The cytochrome P450 enzyme in petunia has flavonoid 3′-hydroxylase (F3′H) activity and affects anthocyanins at the 3′ and the 5′ positions (F3′5′H), resulting in purple or blue flowers [35,36]. In contrast, plants such as roses and carnations, which lack F3′5′H activity, cannot produce purple or blue flowers. We analyzed the expression of differentially expressed genes in three varieties of HPA at three flowering stages. We found that a significant number of genes (such as CHS, ANS, and cytochrome P450s) in the two purple varieties were upregulated during flowering stages, but these genes also showed the same expression trend in white flower varieties or were downregulated in the late flowering stage of purple varieties. Furthermore, we found that a significantly differentially expressed gene, the P450 gene of c120890—which encodes cytochrome P450 monooxygenases, a large group of heme-containing enzymes, and can catalyze NADPH- or NADH-dependent hydroxylation reactions—was significantly upregulated. The P450 gene of c120890 was significantly upregulated (by more than 100 times) in both HC (purple flower) and HS (purple flower) and was barely expressed in HP (white flower) (Figure 2B), indicating that it might be a key regulatory gene of HPA flower color.

The biosynthesis pathway of anthocyanins in plants is well understood [37]. The key enzyme in the bud stage of the whole process is phenylalaninammo-nialyase. Its counterparts in the last two stages are CHS, chalcone isomerase, flavanone-3-hydroxylase, and UFGT, which are the main catalytic enzymes in the conversion of colored but unstable anthocyanins into stable anthocyanins [28]. In our study, the overexpression vectors pCAMBIA3301-CHS and pCAMBIA3301-ANS were transformed into tobacco by using the *Agrobacterium*-mediated method, and the results showed that the flower color of transgenic tobacco deepened, and the contents of anthocyanin and flavonoids increased (Figure 6).

In summary, we detected significant differences in flower color between HPA varieties, and nine transcriptome libraries for the flower color of HPA were constructed. Over 434,349 transcripts were isolated, 302,832 unigenes were identified, and 2098 DEGs were obtained and annotated. Furthermore, color-related DEGs were enriched in 11 pathways, especially in the flavonoid biosynthetic pathway, which is regulated by the cytochrome P450-type monooxygenase gene, which might play a key role in the regulation of the flower color of HPA. RNA-Seq analyses identified 101 gene fragments that were P450 family proteins (29 fragments of P450 family proteins were upregulated, 36 fragments of P450 family proteins were downregulated) (Appendix A). However, the specific regulatory roles of CHS and ANS and the specific proteins in the P450 family that contribute to this process remain to be seen.

## Figures and Tables

**Figure 1 genes-11-00031-f001:**
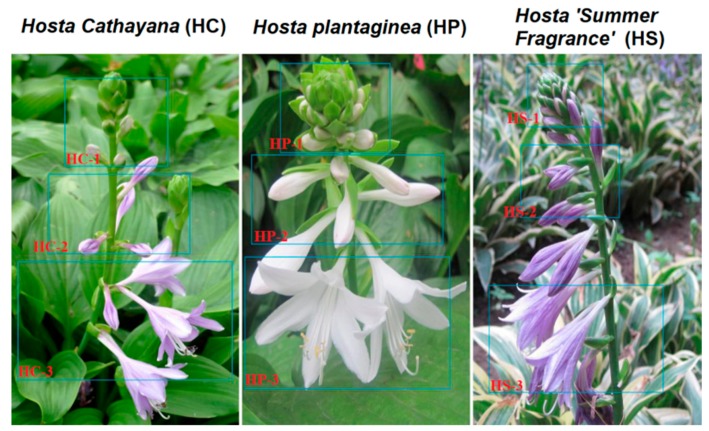
The development and anthocyanin accumulation in *Hosta plantaginea* (Lam.) Aschers. The blue boxes in the images mark the different flowering stages to compare the colors of the three varieties in the flower bud stage (HC-1, HP-1, and HS-1), the initial flowering stage (HC-2, HP-2, and HS-2), and the late flowering stage (HC-3, HP-3, and HS-3).

**Figure 2 genes-11-00031-f002:**
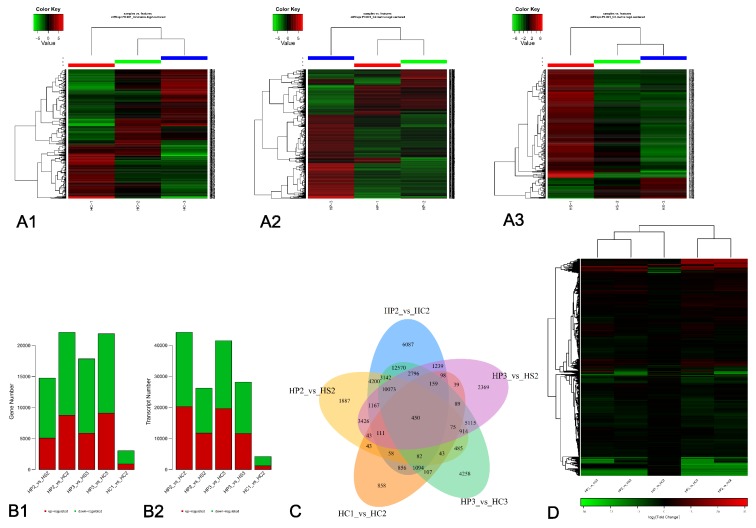
Analyses of differentially expressed genes (DEGs). (**A**) Heatmap of genes differentially expressed in the same plant but at different flowering stages: HC (**A1**), HP (**A2**), and HS (**A3**). (**B**) Comparison of the number of DEGs (**B1**) and differentially expressed proteins (**B2**) between the initial period and the late flowering stages. (**C**) Venn diagram of DEGs for the initial period and the late flowering stages of 450 common genes compared with the initial flowering period and the flowering period. (**D**) Heatmap of genes differentially expressed between the initial flowering period and the flowering period. *p*-value < 0.05 and |FoldChange| > 2.

**Figure 3 genes-11-00031-f003:**
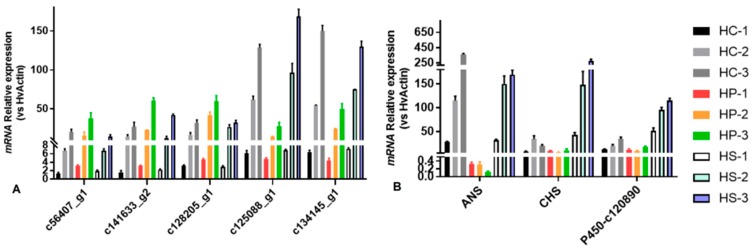
Validation of DEGs by qRT-PCR. (**A**) Five DEGs were randomly selected to verify the accuracy and reliability of the transcriptome sequencing results in the flower bud stage. (**B**) The expression of CHS, ANS, and P450-c120890 in different flowering stages. The expression of actin was used as an internal control. Data represent the mean ± SD of three independent experiments.

**Figure 4 genes-11-00031-f004:**
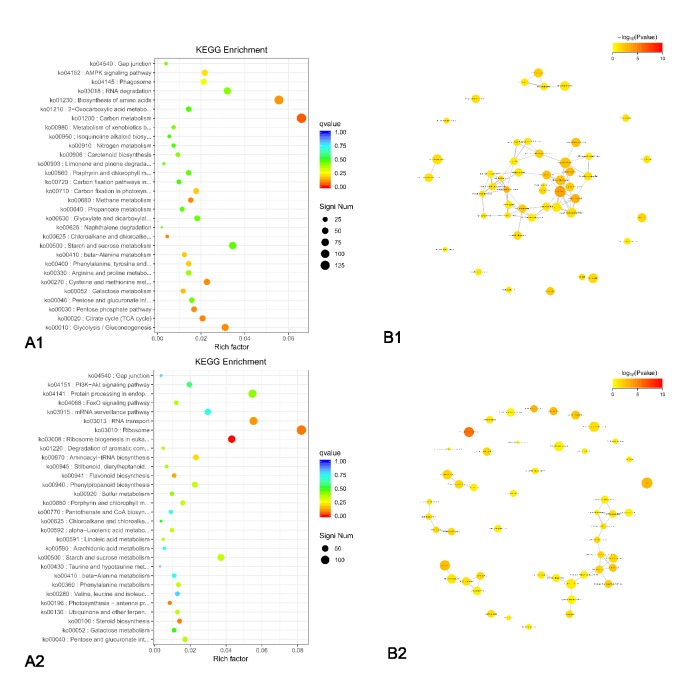
Pathway and functional analysis of DEGs: (**A**) a scatter map of significantly enriched Kyoto Encyclopedia of Genes and Genomes (KEGG) functional terms using clusterProfiler (the color of the nodes represents the enrichment degree of the function (as the *p*-value); the higher the enrichment degree, the lower the *p*-value and the darker the color): (**A1**) DEGs for HP-3 vs. HC-3; (**A2**) DEGs for HP-3 vs. HS-3. (**B**) Functional interaction network of KEGG enrichment terms using the igraph package in R (the metabolism of xenobiotics by cytochrome P450 is marked in red): (**B1**) DEGs for HP-3 vs. HC-3; (**B2**) DEGs for HP-3 vs. HS-3. Green represents downregulation, and red represents upregulation; *p*-value < 0.05 and |FoldChange| > 2.

**Figure 5 genes-11-00031-f005:**
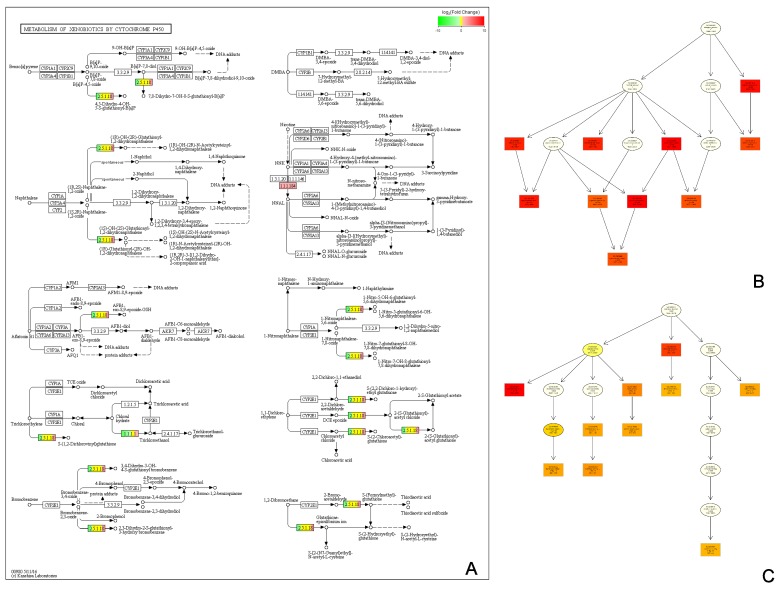
Pathway and functional analysis of the P450 gene. (**A**) The pathway map for the metabolism of xenobiotics by cytochrome P450 was obtained from the statistical analysis of DEG metabolic pathways from KEGG database comparison; (**B**,**C**) the Directed Acyclic Graph (DAG) of the biological process and molecular function for P450-related differential genes was obtained by GO enrichment analysis using topGO. The depth of the color represents the degree of enrichment, with a darker color representing a higher degree of enrichment.

**Figure 6 genes-11-00031-f006:**
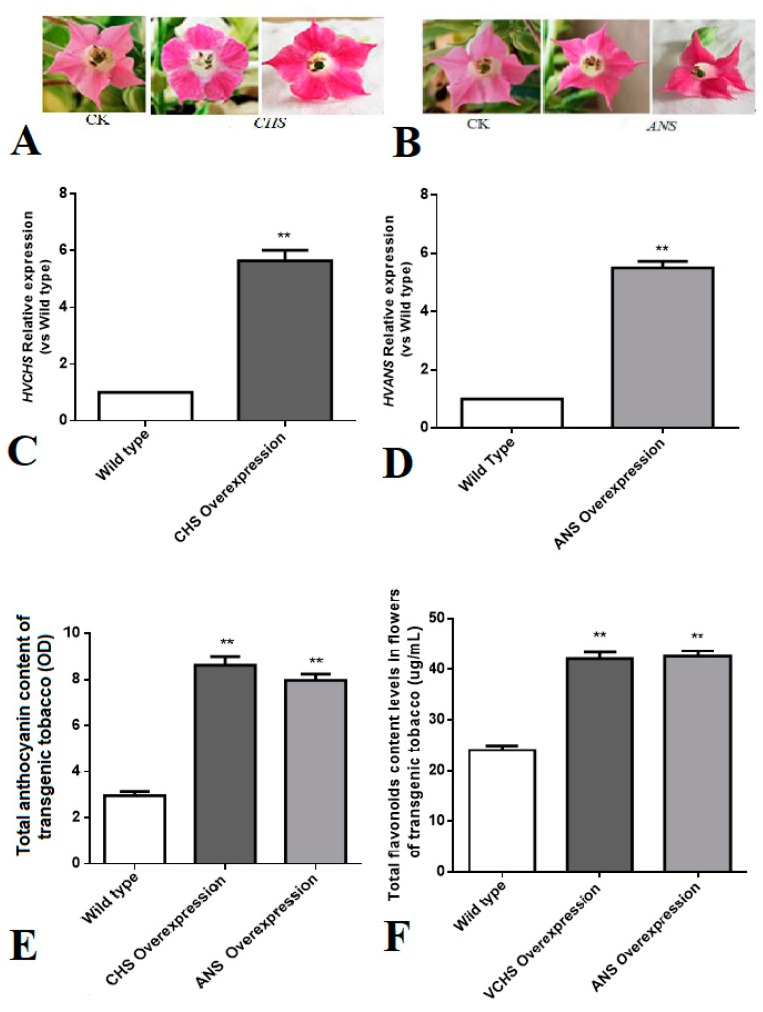
The regulation of flower color by the CHS and anthocyanidin synthase (ANS) genes in tobacco: (**A**,**B**) the color of the transgenic positive CHS or ANS overexpression of tobacco petals was more deepened than CK (the natural color of tobacco petals); (**C**,**D**) the relative expression of the CHS and ANS genes was confirmed to be significantly higher in the overexpressing tobacco flowers compared with wild-type tobacco (CHS gene *p*-value = 0.00194, ANS gene *p*-value = 0.00324); (**E**,**F**) the contents of anthocyanins and flavonoids in flowers of transgenic tobacco overexpressing CHS or ANS increased compared with those in the flowers of wild-type tobacco (anthocyanin *p*-value = 0.00214; flavonoid *p*-value = 0.00414). The relative mRNA expression of CHS and ANS and the increases in anthocyanin and flavonoid content were statistically different compared with wild-type tobacco. The data represent the mean ± SD of three independent experiments. ** *p* < 0.01 vs WT, by One-way ANOVA of GraphPad Prism 7.

**Table 1 genes-11-00031-t001:** Summary of RNA-Seq data.

Sample	Total Sequence	%GC	Sequence Length (bp)
HC-1	48,527,189	47.5	125
HC-2	48,085,043	48	125
HC-3	48,250,194	48.5	125
HP-1	44,116,660	47	125
HP-2	44,512,366	48	125
HP-3	42,504,784	48	125
HS-1	43,869,339	47	125
HS-2	44,195,154	47	125
HS-3	44,053,222	47	125

The transcriptomes of the samples were obtained after processing and assembly. The quantity and quality of the sequencing data are shown in the table. HC: *H. Cathayana*; HP: *H. plantaginea*; HS: *H. plantaginea ‘Summer Fragrance’*.

## Data Availability

The sequencing data by Illumina Hiseq2000 have been uploaded to NCBI BioSample database, No. PRJNA542483. Availability of data and supporting materials section: Please contact author for data requests. Please contact the email: liuhongzhang@jlau.edu.cn.

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
