# Peer review of "Transcriptome-Wide Analysis Reveals Key DEGs in Flower Color Regulation of Hosta plantaginea (Lam.) Aschers"

_genes, 2019, doi:10.3390/genes11010031_

Round 1
Reviewer 1 Report
The main problem of the article is the mixture of the concepts “species” and “variety”. Species is a botanical category. Within one species people create different varieties. Though sometimes Hosta Cathayana is identified as separate botanical species, in reality it is the variety of Hosta plantaginea species, like Summer Fragrance is. So, it is needed to check carefully the text from that point of view (for example line 291 “3 species is not correct, it must be 3 varieties). In addition to that, Latin names of species and other botanical taxa (in this case family name) must be written in italics. Names of varieties must be written in upright font.
Lines 207 – 221. Check English grammar.
Misprints:
Line 68 - ...Diego, CA, USA). technology. Subsequently, the functions...
Line 84 - ... 0.312 %, PH=8.2.These plants...
Line 180 - ... DEGs in HS-3, Other side 21,887 DEGs...
Line 256 - ... the color of the transgenic tobacco plants positive for CHS or ANS overexpression was 256 deepened than CK(The natural color of tobacco petals) and English language.
Line 294 - ... (San Diego, CA, USA).to identify the DEGs...
Author Response
Dear editor and reviewer,
Genes
We really appreciate you and the reviewers for your comments on our Manuscript ID genes-658722 entitled " Transcriptome-wide analysis reveals differentially expressed genes in flower colors of Hosta plantaginea (Lam.) Aschers " We express our sincere gratitude and thankfulness for your time and precision in reviewing our manuscript. The responses to the comments are as follows. For your kind information, we have carefully dealt with the comments of the reviewers as follows: All the places which we changed have already been marked red in our paper, for the purpose of highlight. We hope the revised manuscript meets the standard of publication. Thank you!
Reviewer(s)' Comments to Author:
Reviewer: 1
Comments and Suggestions for Authors
The main problem of the article is the mixture of the concepts “species” and “variety”. Species is a botanical category. Within one species people create different varieties. Though sometimes Hosta Cathayana is identified as separate botanical species, in reality it is the variety of Hosta plantaginea species, like Summer Fragrance is. So, it is needed to check carefully the text from that point of view (for example line 291 “3 species is not correct, it must be 3 varieties). In addition to that, Latin names of species and other botanical taxa (in this case family name) must be written in italics. Names of varieties must be written in upright font.
Thanks for the comment. We have edited according to your suggestion, rechecked all Latin names of species and other botanical taxa (in this case family name) written in italics. Names of varieties must be written in upright font.
Lines 207 – 221. Check English grammar.
Response: Thanks for the comment. We have edited according to your suggestion, the grammar of our manuscript had been rechecked carefully by native speaker, please see line 209-223.
Misprints:
Line 68 - ...Diego, CA, USA). technology. Subsequently, the functions
Response: Thanks for the comment. We have edited, please see the line 68.
Misprints:
Line 84 - ... 0.312 %, PH=8.2.These plants...
Response: Thanks for the comment. We have edited, please see the Line 84.
Misprints:
Line 180 - ... DEGs in HS-3, Other side 21,887 DEGs...
Response: Thanks for the comment. We have edited, please see the Line 182.
Line 256 - ... the color of the transgenic tobacco plants positive for CHS or ANS overexpression was 256 deepened than CK(The natural color of tobacco petals) and English language.
Response: Thanks for the comment. We have edited, please see the Line 258-259.
Misprints:
Line 294 - ... (San Diego, CA, USA).to identify the DEGs...
Response: Thanks for the comment. We have edited, please see the line 296.
Thank you for revising the manuscript

Reviewer 2 Report
In this study, the authors analyse the transcriptome of different flowering stages of three cultivars of Hosta Plantaginea Aschers (HPA) using illumine HiSeq200 technology. The functions and pathways associated with the differentially expressed genes were analysed using Gene Ontology, KEGG and KOG enrichment analyses. Furthermore, Chalcone synthase (CHS) and anthocyanidin synthase (ANS) were overexpressed in transgenic tobacco petals and the relationships between their expression levels and flower color were analysed, suggesting that CHS, ANS, and the cytochrome P450s-regulated flavonoid biosynthetic pathway might play key roles in the regulation of flower color in HPA.
Overall, the manuscript is clearly written, and the work here described contributes new knowledge to the field of the transcriptomic changes and the underlying molecular mechanism of flower color in Hosta plantaginea (Lam.). Some information is missing in Material and Methods and some issues should be clarified, in addition to the correction of minor mistakes, as follows:
Introduction
Line 60: remove the space
Line 67: missing a space
Line 68: remove a point. Diego, CA, USA). technology.
Line 71-73: “Chalcone synthase (CHS) and ANS were overexpressed in transgenic tobacco petals, and the relationships between the expression levels of ANS, CHS, and P450 and flower color were analyzed using qRT-PCR and absorbance values”. In the result section, the authors do not present the expression levels of P450 in transgenic plants nor their relationship with the flower color. This should be reviewed in the new version of the manuscript.
Materials and Methods
Line 82: missing a space. ……perlite=3:1:1,which
Line 84: missing a space and change PH by pH. ……..PH=8.2.These…
Line 90: How much tissue have the authors used to perform RNA extraction from each sample? The authors should clarify whether the flowers collected for this study belong to the same plant or to different plants and how many plants were analyzed.
Line 145: How many transgenic tobacco lines for each gene were generated and analysed? The authors should clarify whether the qRT-PCR data presented in Figure 6C, D correspond to a single transgenic line. Data from at least two or three independent transgenic lines must be submitted.
Results
In Figure 6E, the total anthocyanin content is showed and not anthocyanin expression levels. This must be corrected in the y-axis legend.
Discussion
Line 291-301: This part of the discussion is very repetitive with the results session and should be modified.
-Line 268-270. The sentence should be rewritten. “Flower color is an important quality index of ORNAMENTAL plants, and it directly determines the ORNAMENTAL value of ORNAMENTAL plants”.
-Line 283, 287: correct the space. “..carotenoids, alkaloids, and flavonoids [6,31] .” “colors in flowers[5,6] .”
- Line 294: delete a point. “(San Diego, CA, USA).to identify the DE….
- Line 351: I think that the sentence “RNA-Seq and qRT-PCR analyses identified 101 gene fragments that were P450 family proteins” is not correct. qRT-PCR was used to validate eight differentially expressed gene obtained in the RNA-Seq. This is so?
Author Response
Dear editor and honored reviewers,
Genes
We really appreciate you and the reviewers for your comments on our Manuscript ID genes-658722 entitled " Transcriptome-wide analysis reveals differentially expressed genes in flower colors of Hosta plantaginea (Lam.) Aschers " We express our sincere gratitude and thankfulness for your time and precision in reviewing our manuscript. The responses to the comments are as follows. For your kind information, we have carefully dealt with the comments of the reviewers as follows: All the places which we changed have already been marked red in our paper, for the purpose of highlight. We hope the revised manuscript meets the standard of publication. Thank you!
Reviewer: 2
Comments and Suggestions for Authors
In this study, the authors analyse the transcriptome of different flowering stages of three cultivars of Hosta Plantaginea Aschers (HPA) using illumine HiSeq200 technology. The functions and pathways associated with the differentially expressed genes were analysed using Gene Ontology, KEGG and KOG enrichment analyses. Furthermore, Chalcone synthase (CHS) and anthocyanidin synthase (ANS) were overexpressed in transgenic tobacco petals and the relationships between their expression levels and flower color were analysed, suggesting that CHS, ANS, and the cytochrome P450s-regulated flavonoid biosynthetic pathway might play key roles in the regulation of flower color in HPA.
Overall, the manuscript is clearly written, and the work here described contributes new knowledge to the field of the transcriptomic changes and the underlying molecular mechanism of flower color in Hosta plantaginea (Lam.). Some information is missing in Material and Methods and some issues should be clarified, in addition to the correction of minor mistakes.
Thanks for the comment. We have edited according to your suggestion, rechecked and edited.
Introduction
Line 60: remove the space
Response: Thanks for the comment. We have edited, please see the Line 60.
Line 67: missing a space
Response: Thanks for the comment. We have edited, please see the Line 67-68.
Line 68: remove a point. Diego, CA, USA). technology.
Response: Thanks for the comment. We have edited, please see the line 68.
Line 71-73: “Chalcone synthase (CHS) and ANS were overexpressed in transgenic tobacco petals, and the relationships between the expression levels of ANS, CHS, and P450 and flower color were analyzed using qRT-PCR and absorbance values”. In the result section, the authors do not present the expression levels of P450 in transgenic plants nor their relationship with the flower color. This should be reviewed in the new version of the manuscript.
Response: Thanks for the comment. We performed transcriptome sequencing analysis on 9 samples of three different flowering stages of three varieties of H. plantaginea (white flower), H. Cathayana (purple) and H.'Summer Fragrance' (purple). A total of 434,349 Transcripts were isolated and assembled into 302,832 Unigene, 2098 differential expression profiles were obtained from three flowering stages of H. plantaginea, 722 differential expression profiles were obtained from H. Cathayana, and 606 differential expression profiles were obtained from H.'Summer Fragrance' by gene expression analysis. We had further in-depth analysis we found 84 genes could be identified as putative homologues of color-related genes in two purple species, We found an cytochrome P450 gene,which may encodes a cytochrome P450-type monooxygenase and may be has flavonoid 3',5'-hydroxylase (F3'5'H) activity, This gene is up-regulated in both purple flower varieties with the flowering period, but is hardly expressed in the white flower variety, Due to the fragment length of transcriptome database splicing, we only performed fluorescence quantitative verification on P450, therefore, it is speculated that this gene should play a role in positive regulation of flower color in Hosta plantaginea (Lam.) Aschers.
Materials and Methods
Line 82: missing a space. ……perlite=3:1:1,which
Response: Thanks for the comment. We have edited, please see the line 82.
Line 84: missing a space and change PH by pH. ……..PH=8.2.These…
Response: Thanks for the comment. We have edited, please see the line 84.
Line 90: How much tissue have the authors used to perform RNA extraction from each sample? The authors should clarify whether the flowers collected for this study belong to the same plant or to different plants and how many plants were analyzed.
Response: Thanks for the comment. We have edited, please see the line 90-91.
Line 145: How many transgenic tobacco lines for each gene were generated and analysed? The authors should clarify whether the qRT-PCR data presented in Figure 6C, D correspond to a single transgenic line. Data from at least two or three independent transgenic lines must be submitted.
Response: Thanks for the comment. We have edited, please see the line 146-147.
Results
In Figure 6E, the total anthocyanin content is showed and not anthocyanin expression levels. This must be corrected in the y-axis legend.
Response: Thanks for the comment. We have edited.
Discussion
Line 291-301: This part of the discussion is very repetitive with the results session and should be modified.
Response: Thanks for the comment. We have edited.
Line 268-270. The sentence should be rewritten. “Flower color is an important quality index of ORNAMENTAL plants, and it directly determines the ORNAMENTAL value of ORNAMENTAL plants”.
Response: Thanks for the comment. We have edited, please see the line 270-271.
-Line 283, 287: correct the space. “..carotenoids, alkaloids, and flavonoids [6,31] .” “colors in flowers[5,6] .”
Response: Thanks for the comment. We have edited, please see the line 285,289.
Line 294: delete a point. “(San Diego, CA, USA).to identify the DE….
Response: Thanks for the comment. We have edited, please see the line 68.
Line 351: I think that the sentence “RNA-Seq and qRT-PCR analyses identified 101 gene fragments that were P450 family proteins” is not correct. qRT-PCR was used to validate eight differentially expressed gene obtained in the RNA-Seq. This is so?
Response: Thanks for the comment. We have edited, please see the line 352-354.
Thank you for revising the manuscript
Reviewers:
Remarks: Besides the response to reviewers’ comments, the words or sentences that were changed are highlighted in the premise of ensuring the quality of the article.
Thank you once again.

This manuscript is a resubmission of an earlier submission. The following is a list of the peer review reports and author responses from that submission.
Round 1
Reviewer 1 Report
COMMENTS TO AUTHORS
English hasn’t been revised, and it really need that. There are whole paragraphs that are very difficult to understand. Maybe professional English editing would be desirable.
The author of the species don’t need to be included every time the species is called (e.g. Lines 30,38,57,59,65,68,105, 142,143,149,152x2,153,163,266,314).
As previously suggested, the way the samples are named need to be consistent across the manuscript and stated at the beginning of the manuscript. For example the late flowering stage sometimes is also called full-bloom stage. In any case I suggest to call this third flowering stage (the completely open flower) as anthesis, which is the appropriate term.
Again the three color morphotypes sometimes are considered species (e.g line 16), sometimes varieties (e.g line 27,150,163) and sometimes cultivars (e.g line 59,265). Name them properly. In case they are varieties the first time they are named it should be: H. plantaginea var. plantaginea (white flower), hereafter HP; H. plantaginea var. cathayana (purple flower), hereafter HC; and H. plantaginea var. summerfragance (purple flower), hereafter HS, would be helpful. In case they are cultivars it should be written for example like H. plantaginea ‘Summer Fragance’ or H. plantaginea cv. Summer Fragance.
INTRODUCTION
Lines 18-22: Rewrite this paragraph
Line 50: The first time an abbreviation is used it should be explained. E.g. “under the catalysis of the anthocyanin synthase (ANS)...”
Lines 46-57: Information is repeated. Rewrite.
I still miss information about previous works in Hosta in the introduction
Which are the specific questions of this study? They are not clearly stated in the manuscript and consequently there are no clear answers.
METHODOLOGY
2.1 Plant Materials and growth conditions
Lines 68-71: Rephrase. Better than giving the geographical coordinates of the place, describe the conditions of growing (temperature oscillations, soil composition, etc).
When describing plant materials it is important to make a call to Figure 1, because in that figure is where the reader can see what the authors are describing. Describe here the sample abbreviations for the varieties and for the developmental stages.
Clarify here how the samples were collected: Petals were collected from a single flower? Or they belong to different flowers and/or plants? Similarly, when the different flowering stages of a single variety were collected, did it was at the same time? They belong to the same or different plants?
2.3 Transcriptome assembly
Line 91: Which program was used and how was the criteria established for the high quality standards?
Line 92: Which parameters were used for the assembly in Trimity? The default parameters? If it is, it should be mentioned.
Line 94: Clarify that this expression values come from Trinity.
Line 95: Clarify which three samples
Lines 97-99: I can’t see the link of this paragraph with any previous information. Rephrase
Lines 99-101: Put this info above when talking about Trinity
2.4. Validation of genes expression with qRT-PCR
Line 103: Put this info in the plant materials section instead of here.
Clarify how many technical replicates were used
How were the PCR conditions?
2.5. Gene Ontology annotation
How did the authors determined DEGs? Which program give the Q-value? Which samples are compared? It may be necessary a different section to describe the methodology of the expression analyses (it already have a section in the results). Which programs were used?
2.6 Study on Flower Color Regulation of CHS and ANS Genes in Tobacco
Lines 136-137: Rephrase
It is surprising the section about the transformation experiment with Agrobacterium. Since there are no specific questions in the manuscript, I don’t know exactly what the authors are trying to demonstrate with this experiment. Why hasn’t been included the ANS and CHS genes of HP in the experiment?
RESULTS
3.1. Flower colors of H. plantaginea (Lam.) A. at different developmental stages
Lines 144-147: Rewrite this entire paragraph.
Line 147: “HS varieties is the full-bloom and HC varieties is mauve” I don’t understand what authors want to say.
Lines 150-153: This can be summarized as follows. Flower color in the bud (HC-1, HP-1, and HS-1), initial flowering (HC-2, HP-2, and HS-2) and anthesis stage (HC-3, HP-3, and HS-3) of the three varieties.
3.2. Transcriptome sequencing and assembly
Line 156: Put the info about the program (FastQC) used in the filtering process in Materials and Methods!
Table 1 is not very informative, add the number of reads before and after the filtering process, as well as the number of transcripts assembled and the mean length of them for each sample.
3.3. Analyses of DEGs
Line165: “at the same time” I don’t understand what authors mean. Rephrase
In Figure 2 is almost impossible to see the text of the figures. Use a bigger font size.
I have some concerns, why are only highlighted CHS, ANS and the P450-type monooxygenase genes? Are there any other flavonoid related genes between those DEGs, or only those three? Expression data for the interesting genes (at least those used for the RT-qPCR verification) should be shown somehow, maybe as supplementary material.
3.4. Validation of DEGs by qRT-PCR
Line 185-186: A “regulation pathway” does not exists. Rephrase
Line 186: Instead of flowering stage use anthesis stage or late flowering stage
Line 191: In 3A there are 8 genes, not 11.
Line 194: “Three independent experiment” Add this information to the 2.4 section.
One of the most important results here is that ANS, CHS and P450-c120890 expression values are lower for the white variety than the purples ones. This should be highlighted in text!!! Describe how is the fold-change between samples.
3.5. Pathway and Functional analysis of DEGs
All results description of this section should be rewritten and clarified. It is very hard to follow.
If these analyses were only performed for the late flowering/anthesis stage this should be said in M&M and explain why.
Figures 4 and 5 are completely useless with that size. Is impossible to see something clear. Maybe authors should select for one or two graphs and put the others in supplementary materials.
3.6. Study on Flower Color Regulation of HVCHS and HVANS Genes in Tobacco
Explain what the abbreviation HV means, or delete it in the whole section
Line 225-226: Modify the phrase to something like: The color of the flowers of positive transgenic tobacco plants was deepened
I don’t know exactly what the authors are trying to demonstrate with the transformation experiment. Like it is, this experiment only demonstrate that the expression of CHS and ANS from Hosta is higher than that of wild type tobacco. If CHS and ANS from the white variety (HP) are not included, conclusions about the importance of this genes in anthocyanin production in Hosta cannot be made.
In addition, which is the point of comparing the anthocyanin content between wild-type tobacco and the transformed plants but not including the three wild-type Hosta plants (HC, HP, HS)?
Line 234: 6A is not the vector of Agrobacterium.
DISCUSSION
Discussion has been improved but it still needs huge work. I recommend an extensive literature review.
Line 271: Authors said 84 color related DEGs were found, however I can’t find this data in the results section. Who are these 84 flower color related genes? Why hasn’t been this sequences further studied and compared between colors?
MINOR COMMENTS
In the whole manuscript there are many typographical errors that should be corrected. I recommend authors to carefully read the manuscript to change them. I will only highlight some of them.
Line 11: Hosta plantaginea is not a genus, it is a species. Change the word genus for species.
Line 18: change further more for furthermore
Line 40: a → as
Line 40: which recorded → which is recorded
Line 41: Put the reference of “Compendium of Materia Medica”
Line 46: and other factors → among others.
Line 57: kingds -< kinds
Line127: Change “Seeds of Wild type tobacco NC89, agrobacterium tumefaciens EHA105” for “Seeds of wild type tobacco NC89, Agrobacterium tumefaciens EHA105”
Line 135: were verified → was verified
Line 155: table 1 → Table 1, library → libraries
Line 186: Delete c120890 gene
Line 191: Verify → verify
Line 196: To analyzing → To analyze
Line 198: “we had obtained totally” → “we obtained a total of”
Line 212: P450s protein → P450 proteins
Line 250: varieties → varies
Text in figures 2,4 and 5 is impossible to read. Use a larger font size.
Reviewer 2 Report
The manuscript by Zhang et al. investigates the transcriptome in the flowers of three Hostas varieties at different developmental stages. Different species of Hostas are popular ornamental plants with not only attractive foliage but also with spectacular flowers and in this term the study is justified, in particular that in Hosta plantaginea only two flower colours were observed. Therefore, to breed varieties with other colours would have a great commercial interest too, in addition to answer the question that why this species produces flowers with only two colours.
As a conclusion, the authors state that “The results yield insights to flower color regulation mechanism and contribute great significance for breeding polychrome varieties of ornamental flowers.” line 31-32. Unfortunately, neither of these are true, because they do not explain based on the results that what might be the reason for the only two flower colours and breeding strategies are not proposed either. This is a major lack of the study and has be corrected. To do this I suggest to compare their results with other similar studies, where flowers produce more colour varieties.
Language of the manuscript needs a very serious editing/correction because there are very large number of syntax, grammar and typing errors. In a number of sentences it is impossible to find out what the authors wanted to say. For example: line 13: „which seriously restricts the colour matching application”. Line 168: “By analyzed statistical histogram….”. Line 173: “containing greater concern”. These are just few examples for the bad English of the manuscript.
Specific remarks
Be consistent with the naming of the flowering stages. Bud, initial and late flowering stages, as described in the introduction, would be fine throughout the manuscript. How CHS, ANS, and cytochrome P450 are involved in the development in the two colour system? In particular ANS should have a central role due to its position in the anthocyanin biosynthesis, but that has to be explored and described. Is the two colour system is a general phenomenon in the genus or only appear in plantaginea? It has to be described. Is the white flower plant a separate variety or it is the wild type? Use the correct nomenclature for the varieties names throughout the entire manuscript. Description in Figure 1 is not correct. For example, H. Cathayana is wrong, it has to be Hosta ‘Cathayana’, or even more precisely (because there are more species) Hosta plantaginea ‘Cathayana’ or H. plantaginea ‘Cathayana’. The same for Summer Fragrance. The variety name is NEVER in italics. Visual examination of flower colour (lines 144-145) is very subjective and the anthocyanin content has to be measured in all developmental stages, as it was measured for the transgenic tobacco. There is a general problem with the figures that lettering is too small in size and resolution, cannot be read. Also, where colour scale is given, it has to be described in the figure legend what it represents. Line 169-171. This description is not clear, has to be rewritten. Not clear what the authors wabted to say. It is not HP-3 to HS-3, it is “versus”. Figure 1 and associated text: As I mentioned before figure lettering is very small, so I could not figure out what is the meaning of the red and the green colour on the heatmap. If they are up and downregulation, then this has to be in comparison between two stages, but on the figure it looks like only 1 stage is presented. Venn is not a map, it is a diagram. Line 179: they say that the comparisons between initial and late stages, but HC1 is a bud stage. Why certain stages were compared, but not others? It has to be justified why only that three genes, CHS, ANS and P450-type 174 monooxygenase genes were selected out of the 450 DEGs for further analysis. Part 3.3 has to be intensively modified, corrected, and rewritten. Part 3.4. Figure 3A. The authors say: ‘Random selection of 11 differentially expressed genes to Verify the accuracy and reliability of transcriptome sequencing results in the flower bud stage.” Verify is a typing error. Why the bud stage was chosen for validation, if this stage was not intensively included into DEG comparisons? The result only shows the level of mRNA in a stage and this does not validate the transcriptome expression. A validation has to be comparing the two. Because this cannot be done at the absolute level, it has to be done by comparing two stages and then make a correlation between the RT-PCR and the transcriptome. Figure 3B: ANS is clearly downregulated in the white flower variety and this might be the explanation for the white colour. The authors have to figure out how this can be and write it down. Figure 4. Lettering is too small again, the figure cannot be evaluated. The figure has to be reconstructed and the associated text has to be rewritten. For example “site” is the wrong world in that context. Figure 5 cannot be evaluated again. Part 3.6.: incorrect wording and syntax errors. 3 times has to 3-fold higher and so on. Figure 6A: vector of agrobacterium is meaningless. Discussion: bad language, has to be rewritten.
Reviewer 3 Report
In general presented article is actual and informative. Authors have used different modern techniques and obtained interesting results. But the discussion of these results can be improved by correct comparison of the data received for different evaluated genotypes. The problem is the incorrect use of botanical nomenclature. Firstly, genus Hosta according to the last classification belongs to the family Hostaceae see: Czerepanov S.K. Vascular Plants of Russia and Adjacent States (the former USSR). Cambridge University Press, 1995. 516 p. But it is not a real problem. The problem is that authors don’t separate different terms: species and varieties of the same species, properly. As I understood, they used 2 species: Hosta cathayana and Hosta plantaginea (represented by unnamed varieties) and Summer Fragrance variety of Hosta plantaginea species. This fact is very important, because the degree of relatedness between evaluated genotypes must be taken into account while discussing the results. It was not pointed in the discussion.
Also, English grammar must be improved.